# Submolecular probing of the complement C5a receptor–ligand binding reveals a cooperative two-site binding mechanism

Andra C. Dumitru[1,4], R. N. V. Krishna Deepak[2,4], Heng Liu[3,4], Melanie Koehler [1], Cheng Zhang[3✉], Hao Fan [2✉] & David Alsteens [1✉]

A current challenge to produce effective therapeutics is to accurately determine the location of the ligand-biding site and to characterize its properties. So far, the mechanisms underlying the functional activation of cell surface receptors by ligands with a complex binding mechanism remain poorly understood due to a lack of suitable nanoscopic methods to study them in their native environment. Here, we elucidated the ligand-binding mechanism of the human G protein-coupled C5a receptor (C5aR). We discovered for the first time a cooperativity between the two orthosteric binding sites. We found that the *N*-terminus C5aR serves as a kinetic trap, while the transmembrane domain acts as the functional site and both contributes to the overall high-affinity interaction. In particular, Asp282 plays a key role in ligand binding thermodynamics, as revealed by atomic force microscopy and steered molecular dynamics simulation. Our findings provide a new structural basis for the functional and mechanistic understanding of the GPCR family that binds large macromolecular ligands.

[1] Université catholique de Louvain, Louvain Institute of Biomolecular Science and Technology, 1348 Louvain-la-Neuve, Belgium. [2] Bioinformatics Institute (BII), Agency for Science, Technology and Research (A*STAR), Singapore, Singapore. [3] Department of Pharmacology and Chemical Biology, School of Medicine, University of Pittsburgh, Pittsburgh, PA, USA. [4] These authors contributed equally: Andra C. Dumitru, R. N. V. Krishna Deepak, Heng Liu. ✉email: chengzh@pitt.edu; fanh@bii.a-star.edu.sg; david.alsteens@uclouvain.be

G-protein-coupled receptors (GPCRs) are a large family of versatile and dynamic cell surface receptors expressed in almost all tissue types. Upon ligand binding to highly conserved orthosteric binding sites, GPCR pass from a quiescent to an active state, initiating signal transduction. The conservation of orthosteric binding sites throughout the GPCR superfamily and a poor understanding of the mechanism underneath their activation cycle are two key factors currently hampering the development of specific drugs towards a specific sub-type in a GPCR family. The G-protein-coupled receptor C5a anaphylatoxin chemotactic receptor 1 (C5aR) belongs to the rhodopsin family of seven transmembrane-containing GPCRs and has been a topic of interest in the last couple of decades due to its relevance in several inflammatory pathologies, such as asthma, arthritis, sepsis, and more recently Alzheimer's disease and cancer[1–3]. Complement C5a anaphylatoxin binding and activation of C5aR elicits a variety of immunological responses in vivo[4], such as chemotaxis, cell activation, and inflammatory signaling. Despite the fact that the interaction between C5a and C5aR is of considerable therapeutic value, their molecular binding mechanism remains elusive. Previous reports suggest, through indirect evidence, a two-site binding mechanism: (i) a binding site, where the C5a rigid core interacts with the N-terminus and the second extracellular loop (ECL2) of the receptor[5] and (ii) an effector site, where the flexible C-terminal fragment of C5a interacts with the cavity formed by the seven transmembrane (7-TM) helices of C5aR (Fig. 1a). The effector site is thought to be responsible for the functional activation of the receptor. On one hand, the main interactions at the binding site occur between C5aR sulfonated tyrosine residues (Y11 and Y14) and C5a residues including R40, R37, and possibly H15[5–7]. On the other hand, the C5a R74 residue is pointed out in several studies as critical for effector site

binding[5,7]. Recently, a structural study revealed the orthosteric action of PMX53, a cyclic antagonist peptide that mimics the C-terminal structure of the endogenous C5a ligand, as well as its effect of stabilizing the C5aR structure (Fig. 1a)[8]. PMX53 shows nanomolar affinity towards the effector site, interacting through several residues including Asp-282 and Arg-175, though leaving the sulfonated amino acid residues at the binding site exposed[8].

However, given the absence of the structure of the C5a–C5aR complex, a full characterization of their interactions is currently missing. Moreover, there is no molecular evidence of the kinetic and thermodynamic contributions of the different binding sites to the overall receptor–ligand binding. It is still unclear whether the effector site and binding site are acting in concerted manner or separately. Understanding this process is likely to illuminate the binding paradigm common to members of the GPCR family that bind large macromolecular ligands. Here, we unravel key elements of the binding mechanism of the C5a–C5aR complex at submolecular resolution. We use force–distance curve-based atomic force microscopy (FD-based AFM), a well-established method enabling to image and probe biological systems at nanometer resolution and to extract their nanomechanical properties (e.g., topography, adhesion, elasticity, and stiffness)[9–13]. Functionalization of the AFM tip with specific ligands targeting C5aR enables the detection of (bio-)chemical interactions while extracting their structural, thermodynamic, and kinetic parameters. For the first time, we decipher the binding properties of a single ligand towards its cognate receptor at unprecedented sensitivity. Our approach enables a quantitative analysis of the binding mechanism of a large ligand at the submolecular level, revealing a combination of kinetic and thermodynamic insights into the ligand-binding process. Together with steered molecular dynamics (SMD) simulation and functional assays, our

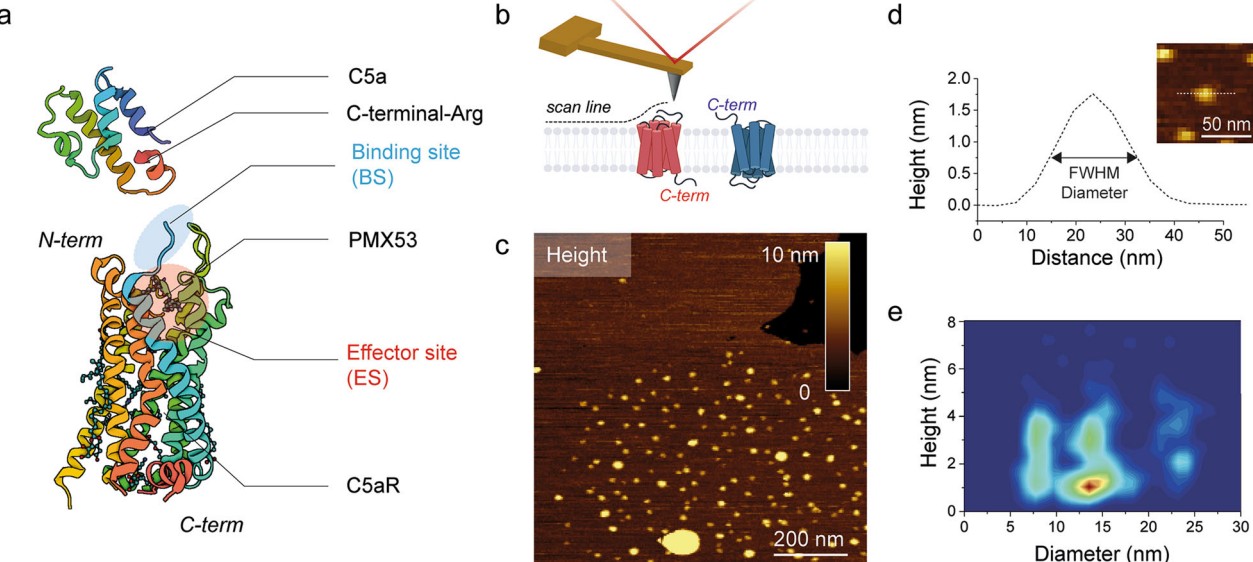

**Fig. 1 FD-based AFM mapping of C5aR receptors and probing their orientation within the lipid bilayer. a** Ribbon diagrams of human C5aR and C5a structures. The interaction between C5a and C5aR is stabilized by two orthosteric binding sites at the receptor extracellular side: a binding site at the N-terminus (shaded in blue) and a functionally important effector site at the extracellular region (shaded in red) of C5aR. The peptide antagonist PMX53 establishes hydrogen bonds with ECL2 at the effector site. The Arg (R) residue at the C-terminal of the C5a ligand is thought to play a key role in stabilizing the interaction with C5aR. **b** Orientation of lipid bilayer-embedded C5aR is random: they can adopt two orientations, with the intracellular C-terminal His$_6$-tag facing the inner or the outer side of the lipid bilayer. **c** Overview AFM topography image (height map) of C5aR reconstituted in liposomes and adsorbed on freshly cleaved mica. Sparsely distributed C5aR particles can be observed protruding from the liposomes. The image was acquired with a bare AFM tip. **d** Cross-section (white dashed line in inset) showing a C5aR particle protruding 1.7 nm from the lipid bilayer having a diameter of 16 nm. The diameter was measured as full-width at half-maximum (FWHM). Inset: expanded view of a single C5aR particle. **e** 2D histogram of height and diameter of C5aR receptors imaged in (**c**). The diameter distribution shows three main populations, while the height distribution shows two main peaks. Data in (**c**) and (**e**) are representative of at least five independent experiments.

observations point toward a cooperative model in which the binding site acts as a kinetic trap, effectively boosting local ligand concentration and promoting the interaction at the functionally relevant effector site. From a broader perspective, this study provides new insights on the role of positive allosteric interactions from a kinetic, thermodynamic, and functional point of view.

## Results

**C5aR adopts random orientations after reconstitution in lipid membranes**. Reconstitution of C5aR into proteoliposomes was confirmed by western blot and the band at ≈45 kDa is in good agreement with the expected size of 43 kDa calculated from the C5aR sequence (Fig. S1). Before characterizing the binding properties of C5a to C5aR, we imaged purified C5aR reconstituted in liposomes and adsorbed on freshly cleaved mica in buffer solution using FD-based AFM. While membrane receptors embedded in their native cellular membrane always have a unique orientation, they lose this original orientation through the reconstitution steps into liposomes[10]. Within lipid bilayers, embedded receptors can adopt both orientations, having either their extracellular or intracellular side exposed to the AFM tip (Fig. 1b).

To detect the C5aR orientation within the lipid bilayer, we first imaged our sample at high-resolution using FD-based AFM to discriminate topographical characteristics corresponding to one of the two orientations. In our instrumental FD-based AFM setup, an oscillating AFM cantilever is continuously approached and retracted from the sample surface in a sinusoidal manner. The sample topography along with the adhesion can be simultaneously extracted from each FD curve[9,13,14]. We imaged the sample with a bare AFM tip and observe sparsely distributed C5aR particles protruding away from the lipid bilayer surface (Fig. 1c). The height of the emergent part of C5aR above the lipid bilayer (protrusion height) as well as its diameter (calculated as the full-width at half-maximum) were extracted for each particle and plotted in a two-dimensional height–diameter histogram (Fig. 1d, e). A bimodal distribution can be observed for the heights, with two peaks centered, respectively, at $1.1 \pm 0.5$ and $2.8 \pm 1.2$ nm, while the presence of three main populations can be observed for the diameters, with peaks centered at $8.6 \pm 0.4$, $14.1 \pm 0.3$, and $23.4 \pm 1.2$ nm (Fig. S2). The two peaks of height distribution could represent the heights of extracellular and intracellular regions of C5aR protruding from the DOPC/CHS membrane. The three peaks in the diameter distribution could be attributed to the presence of C5aR dimers and higher-order oligomers[8,14,15]. For monomers, we clearly observed that both orientations are present without preference (similar density in the 2D histogram).

**Identification of C5aR orientation using affinity imaging**. After showing that we can distinguish between the two possible receptors' orientations via their protrusion heights, we combined topography and affinity imaging using functionalized AFM tips to identify C5aR intracellular and extracellular sides (Fig. 2). Silicon tips were functionalized with a poly(ethylene glycol) linker (PEG), followed by grafting of tris-N-nitrilotriacetic acid (tris-NTA) molecules. The individual tetradentate NTA ligand forms a hexagonal complex with $Ni^{2+}$ ions, leaving two remaining binding sites accessible to electron donor nitrogen atoms from the histidine sidechains of the His8-tag engineered at the terminal end of a polypeptide. To specifically probe one side of the C5aR, we either used a tris-NTA–$Ni^{2+}$ tip to target the intracellular side using the His8-tag present at the C5aR C-terminal end (Fig. 2a–d), or we further derivatized the tris-NTA–$Ni^{2+}$ tip with the endogenous C5a ligand to specifically probe the interaction

with the ligand-binding site at the C5aR extracellular side (Fig. 2e–h). The C5a ligand had a His6-tag engineered at its N-terminal end, which allowed its tethering to the tris-NTA–$Ni^{2+}$ tip. To show the capabilities of this new multiplex approach, we probed the same lipid patch using both C5a and tris-NTA–$Ni^{2+}$ AFM tips. Adhesive events were considered to be specific if they were detected at tip-sample distances of $10 \pm 5$ nm, corresponding to the length of the extended PEG linker, and when the adhesion force was at least two times higher than the noise level (measured at the baseline of the retraction curve). Additionally, each specific adhesion event was validated by fitting the extension profile of the PEG linker using the worm-like chain model[16]. Representative FD curves are presented in Fig. 2d, h showing either specific adhesion events (curves 1 and 2) or unspecific/no interaction FD curves (curves 3 and 4). Control experiments using bare tips or an amino-derivatized tip show either no interaction or unspecific adhesion events (Fig. S3a–d). Finally, blocking experiments using free C5a in solution or injection of EDTA significantly reduces the binding probability, confirming the specificity of both probed interactions (Fig. S3e–h).

For the two types of tip functionalization, the FD curves showing specific adhesion events were analyzed and the interaction force was extracted, as well as the height of the C5aR on which the FD curve was recorded. These values were displayed in the form of 2D histograms of force as a function of height (Fig. 2i, j). We observed that tris-NTA–$Ni^{2+}$ functionalized tips mostly interact with receptors protruding higher from the lipid bilayer ($3.1 \pm 1.0$ nm), while C5a tips were found to interact specifically with lower receptors ($1.7 \pm 0.5$ nm). Together, these results confirm that the receptor orientation can be determined using only their protrusion heights as those are significantly different (Fig. 2k). An overlay of the AFM topography and the specific adhesion events recorded (colored pixels) on the same area with both functionalized tips (C5a tip in red and tris-NTA–$Ni^{2+}$ tip in blue) reveals the identity of the side exposed to the tip (Fig. 2l). Receptors having a protrusion height under a threshold of 1.75 nm were encircled. Together, these data confirm the possibility to identify with a high-probability (> 95%) the receptors oriented in their native state. Therefore, this criterion will be further used in force spectroscopy experiments to validate our measurements.

**PMX53 binds to C5aR with high-affinity**. The lack of a crystal structure characterization of C5a binding to C5aR has impeded a better understanding of the molecular mechanism of action of various C5a ligands. Recently, we provided the high-resolution structure of C5aR with PMX53 and uncovered the orthosteric action of the antagonist[8]. Once able to precisely identify C5aR orientation within the lipid membrane, we decided to study the dynamics of PMX53 binding to C5aR. FD-based AFM and pulling simulations were used to quantify the free-energy landscape of the PMX53–C5aR interaction and the role of the key residues in the binding process (Figs. 3 and 4).

To this end, in FD-based AFM experiments we tethered the high-affinity PMX53 antagonist to the AFM tip and then measured its interactions with C5aR (Fig. 3a). We simultaneously recorded FD-based AFM height images and adhesion maps (Fig. 3b, c) and extracted FD curves located on C5aR having their native orientation (based on our height criterion) (Fig. 3d). Generally, force-probing methods such as FD-based AFM measure the strength of single bonds under an externally applied force. Described first by the Bell-Evans model[17], an external force stressing a bond reduces the activation-energy barrier toward dissociation and, hence, reduces the lifetime ($\tau$) of the ligand–receptor pair (Fig. 3e). The model predicts that

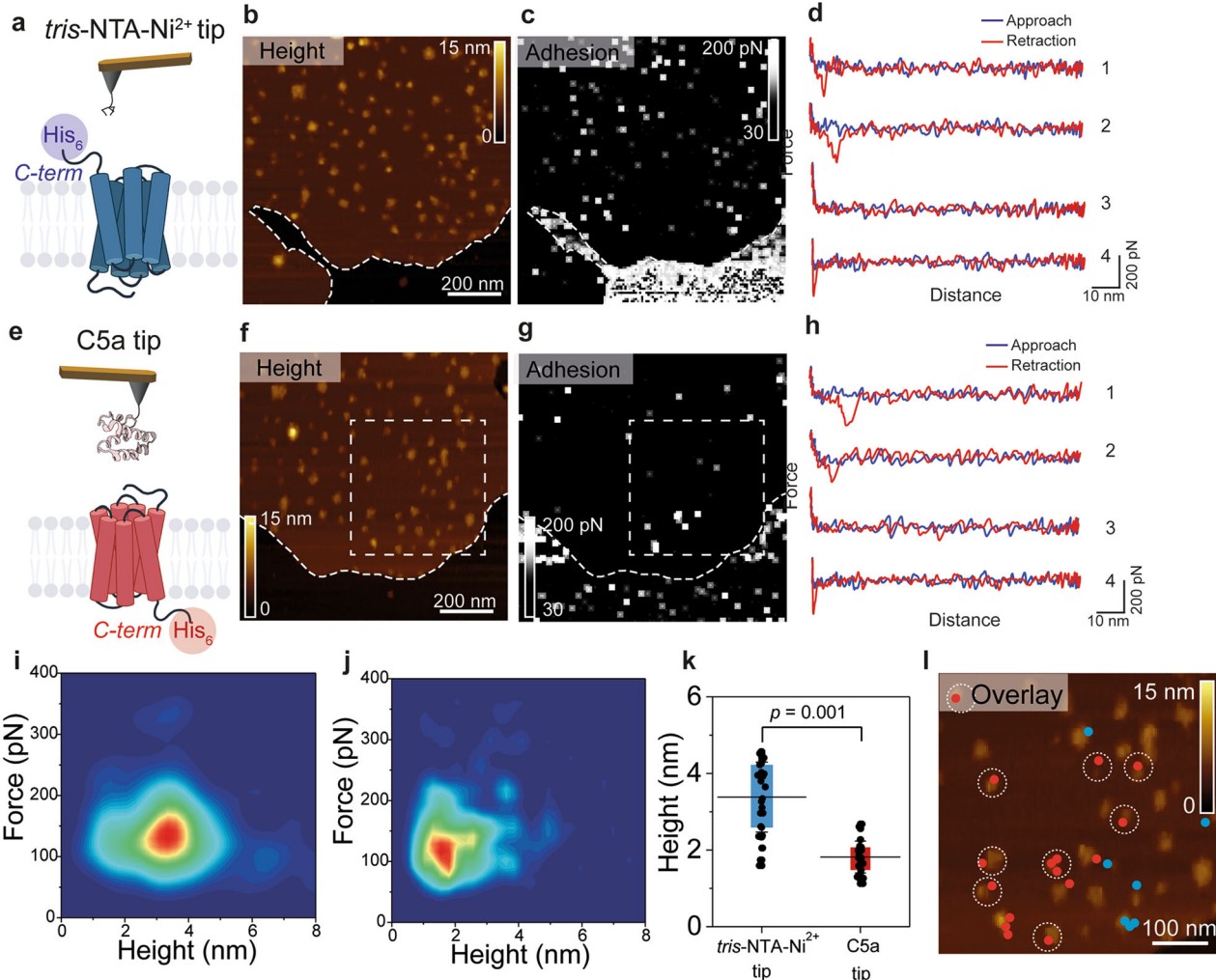

**Fig. 2 Multiplex probing of C5aR intra- and extracellular binding sites as a method discriminate C5aR orientation within lipid membrane.** Two different AFM tip chemistries were used to target either the His$_6$-tag C-terminal end of C5aR using *tris*-NTA–Ni$^{2+}$ functionalized AFM tips (**a–d**) or the N-terminal end of C5aR using the endogenous C5a ligand (**e–h**). AFM height and adhesion images were recorded over the same lipid patch with a *tris*-NTA–Ni$^{2+}$ tip (**b**) and (**c**) and a C5a ligand tip (**f**) and (**h**). **d, h** Representative FD curves showing either specific adhesion events (curves 1, 2) or no/unspecific interactions (curves 3, 4) were extracted from the adhesion maps in (**c**) and (**g**). 2D histograms of force vs. height for *tris*-NTA–Ni$^{2+}$ modified tips (**i**) and C5a tips (**j**). **k** Height distribution of the receptors interacting with the *tris*-NTA–Ni$^{2+}$ or the C5a tip. Two populations can be clearly distinguished, one below 1.75 nm in height, where C5a tips mostly interact with the extracellular side of C5aR, and another one above 3.5 nm in height, where *tris*-NTA–Ni$^{2+}$ functionalized tips interact with the intracellular side of C5aR. **l** Height map overlay of the region marked by a white square in (**f**) and corresponding specific adhesion events extracted from the same areas in the maps in (**c**) and (**g**). Adhesion events between the C5a ligand and the N-terminal side of C5aR are shown as red dots, while the events rising from the *tris*-NTA–Ni$^{2+}$ AFM tip interaction with the His$_6$-tagged C-terminal side of C5aR are displayed as blue dots. White dotted circles mark receptors with a height less than 1.75 nm, where the C5a ligand and the N-terminal side of C5aR interact. The overlay image shows how the orientation of single C5aR particles can be identified using our multiplex probing method. Data are representative of at least three independent experiments. Data in (**k**) are displayed as mean ± S.D. and the ANOVA one-way Tukey test was used to report the statistical significance.

far-from-equilibrium, the rupture force (e.g., binding strength) of the ligand–receptor bond is proportional to the logarithm of the loading rate (LR), which describes the force applied over time. Recently, Friddle, Noy, and de Yoreo (FNdY) introduced a model to interpret the nonlinearity of the rupture forces measured over a wide range of LRs and suggested that this nonlinearity arises through the re-formation of bonds at small LRs[18]. This model provides direct access to the equilibrium free energy $\Delta G_{bu}$ between bound and unbound states (see Methods). The nonlinear oscillating approach and retraction movement of the AFM tip with respect to the sample results in a wide variety of velocities explored during the rupture of the bonds established between the PMX53 derivatized tip and the C5aR (Fig. 3f). To further increase the range of velocities explored, we combined the force–volume

(FV) mode at low speed to reach low LRs and FD-based AFM to explore unbinding process at high LRs (Fig. 3f). For each FD showing a specific adhesion event, we extracted the binding force and the LR measured as the slope of the force vs. time curve just before the rupture (Fig. 3g). When plotting the resulting binding forces as a function of the LRs (also called dynamic force spectroscopy (DFS) plot) on a semi-logarithmic scale (Fig. 3h), a nonlinear dependency of the force with the loading rate is observed as predicted by the FNdY model[18]. Using this model, we extracted the equilibrium force $F_{eq}$, as well as thermodynamic and kinetic parameters such as the binding equilibrium free energy $\Delta G_{bu}$ and the receptor–ligand half-life $\tau_{0.5}$. The dissociation constant $K_d$ was calculated using the relation $\Delta G_{bu} = k_b T \times \ln (0.018 K_d)$, with $0.018 \, l \, mol^{-1}$ being the partial molar volume of

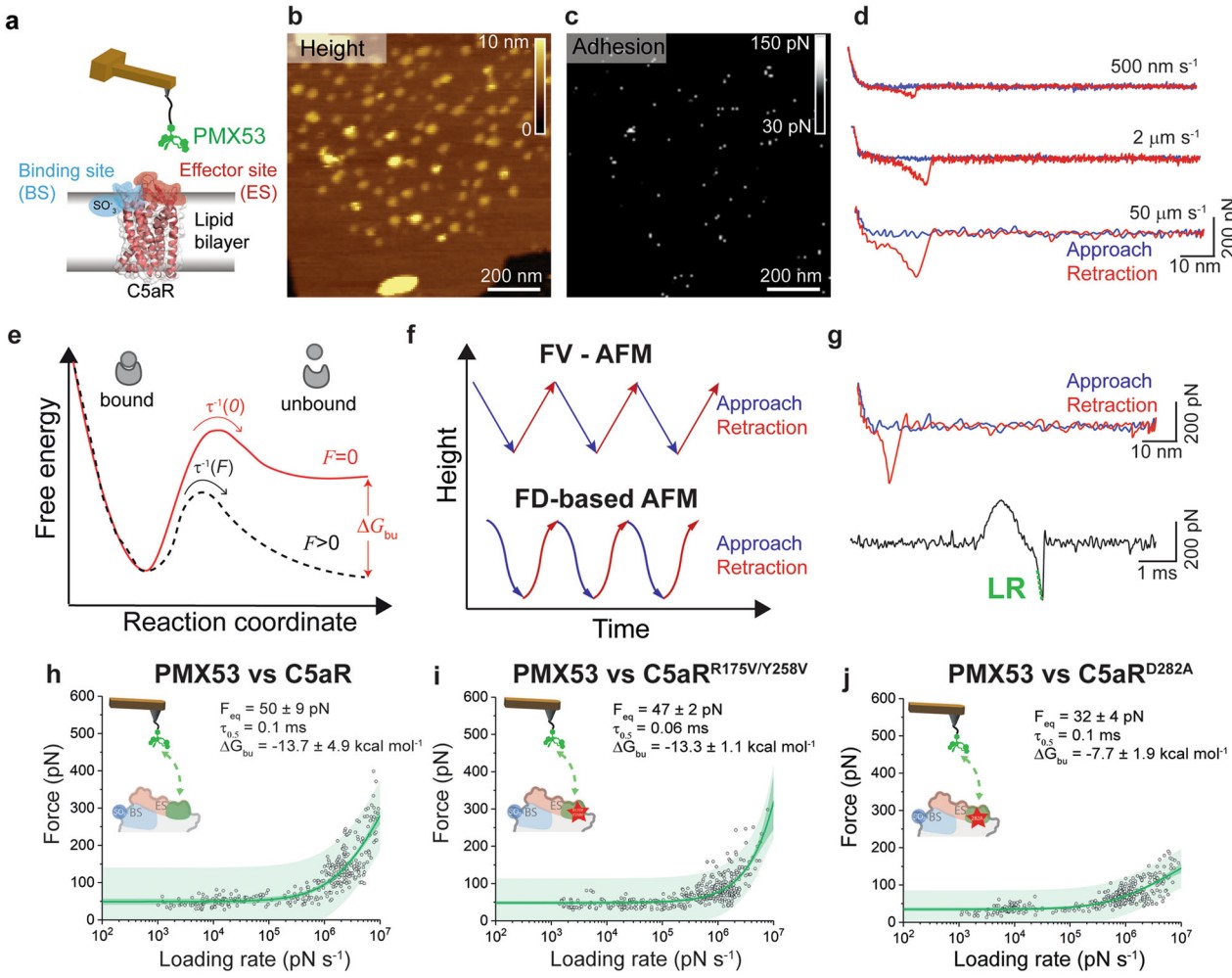

**Fig. 3 Probing the kinetic and thermodynamic parameters underlying PMX53 antagonist binding to C5aR. a** Schematic representation of an AFM tip tethered with the high-affinity PMX53 antagonist probed against C5aR. Height (**b**) and adhesion (**c**) maps recorded while probing C5aR embedded in the lipid bilayer with a PMX53 modified AFM tip. **d** The interaction between PMX53 and C5aR was probed over a wide range of LRs by variating the retraction speed in the force–distance curves. Low LRs were explored at 500 nm s$^{-1}$ and 2 µm s$^{-1}$ pulling speeds, while high LRs were reached at 50 µm s$^{-1}$ pulling speed. **e** Extracting the parameters describing the PMX53–C5aR free energy landscape. A ligand–receptor bond can be described using a simple two-state model, where the bound state resides in an energy valley and is separated from the unbound state by an energy barrier. The transition state must overcome an energy barrier to separate ligand and receptor. $\tau^{-1}(F)$ and $\tau^{-1}(0)$ are residence times linked to the transition rates for crossing the energy barrier under an applied force $F$ and at zero force, respectively. $\Delta G_{bu}$ is the free-energy difference between bound and unbound state. **f** Force–volume (FV)-AFM and FD-based AFM were used to explore binding at low and high LRs, respectively. For each pixel of the topography, the tip is approached and retracted using a linear (FV-AFM) or oscillating movement (FD-based AFM). **g** A force–distance curve (upper panel) can be displayed as a force–time curve (bottom panel), from which the loading rate can be extracted via the slope of the curve just before bond rupture. Probing the kinetic and thermodynamic parameters underlying PMX53 antagonist binding to C5aR (**h**) and and two mutants, (**i**) C5aR$^{R175V/Y258V}$ and (**j**) C5aR$^{D282A}$. Fitting the data using the Friddle–Noy–de Yoreo model (thin green lines) provides average $F_{eq}$, $\Delta G_{bu}$, and residence time ($\tau_{0.5}$) values with errors representing the s.e.m. Each circle represents one measurement. Darker green shaded areas represent 99% confidence intervals, and lighter green shaded areas represent 99% of prediction intervals. A reduction of the affinity WT > R175V/Y258V > D282A is observed. For each condition, data are representative of at least three independent experiments.

water. Fitting the experimental data with the FNdY model provided an equilibrium force $F_{eq}$ of 50 ± 9 pN corresponding to a binding equilibrium free energy $\Delta G_{bu}$ of −13.7 ± 4.9 kcal mol$^{-1}$ for PMX53–C5aR interaction. This value is very similar to the value determined by the SMD simulation for the PMX53–C5aR (−13.8 kcal mol$^{-1}$, see next section). The PMX53 affinity constant towards C5aR, $K_d$ of 4.7 nM, is in good agreement with previous studies where values between 1 and 50 nM were found depending on the species and cell type[19,20]. The high-affinity in the nanomolar range is a result of the PMX53 reduced size and the various substitutions (compared to the C5a native C-terminus) that maximizes the number of interactions within the binding pocket, stabilizing the structure of the C5aR.

**Steered molecular dynamics of C5aR–PMX53 complex.** To get more insight into the PMX53–C5aR binding dynamics we performed molecular dynamics (MD) and steered MD (SMD) simulations (Fig. 4). Over the course of the 300 ns unrestrained MD simulation, the C5aR structure remained stable with no significant structural changes as evidenced from the root-mean-square deviation (RMSD) profile (Fig. S4a). Apart from the N- and C-terminal regions of C5aR, most of the structural fluctuations were observed primarily in the ECL2 and the terminal regions of TM5, TM6, and TM7, while the 7-TM core of the protein remained fairly stable (Fig. S4b). No major structural rearrangements were observed with respect to the PMX53 molecule, which remained tightly bound to C5aR, forming several

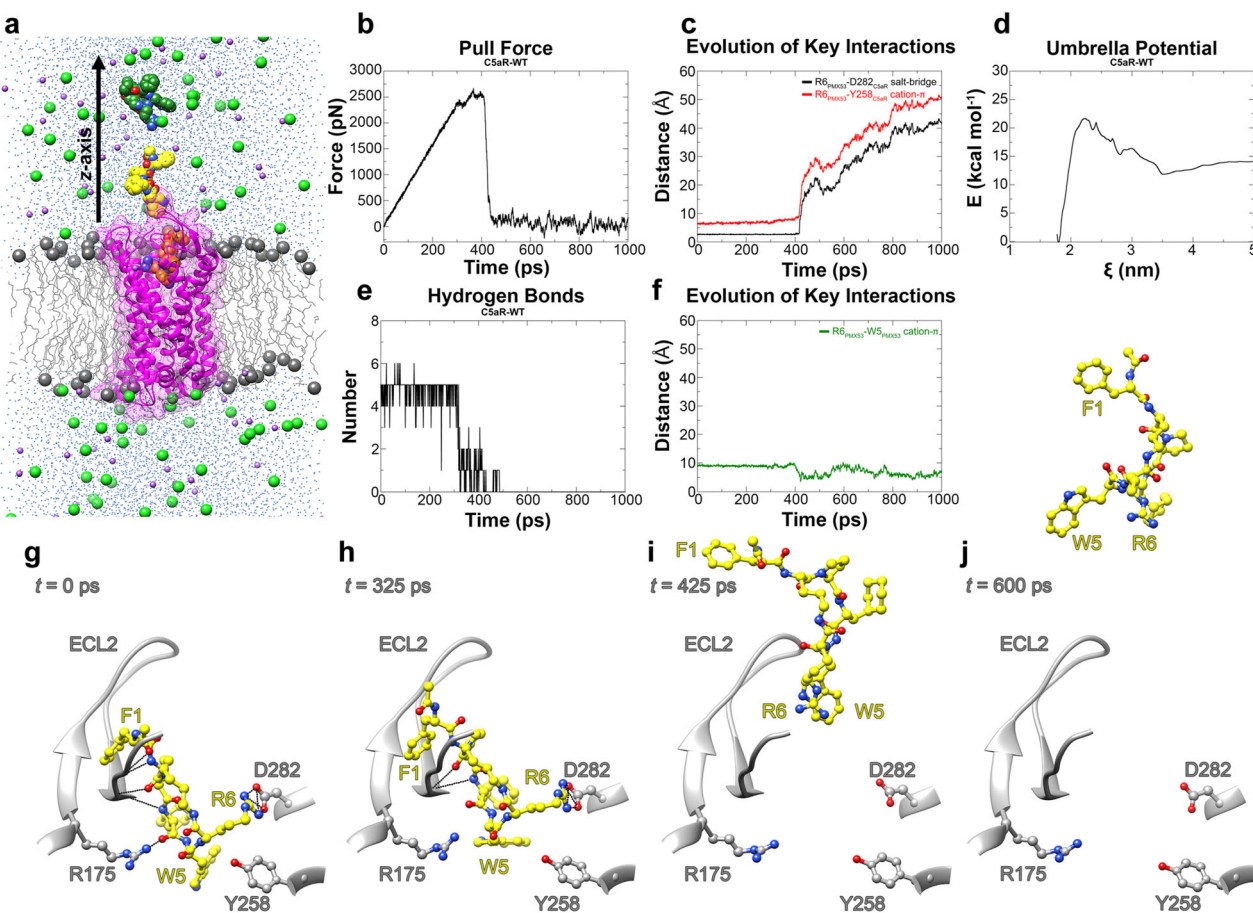

**Fig. 4 Steered molecular dynamics (SMD) or center-of-mass (COM) pulling simulation of C5aR (WT)–PMX53 complex. a** Cut-through section of C5aR (WT)–PMX53–POPC system used for equilibrium MD and steered MD simulations. C5aR is shown in ribbon representation (magenta), embedded in a POPC bilayer (gray) with the headgroup phosphorous atoms shown in sphere representation and the rest of the lipid molecules shown in wire representation. TIP3P water molecules are colored blue, $Na^+$ ions purple, and $Cl^-$ ions green. Positions and conformations of PMX53 at $t = 0$ ps, $t = 500$ ps, and $t = 1000$ ps derived from the COM pulling simulation are shown in orange, yellow, and dark green colors, respectively. The black arrow is along the z-axis and indicates the direction of pulling of PMX53. **b** Plot showing force (pN) vs. time (ps) profile obtained for the C5aR (WT)–PMX53 system with a pulling rate of 5 nm ns$^{-1}$. **c** Evolution of key intermolecular interactions between C5aR (WT) and PMX53, namely the R6PMX53–D282C5aR salt-bridge (black), and the R6PMX53–Y258C5aR cation-π interaction (red) over the course of the pulling simulation. **d** Potential of mean force profile calculated for the dissociation of PMX53 from C5aR (WT) using WHAM following umbrella sampling simulations for the C5aR (WT)–PMX53 system. The average PMF profile calculated using bootstrap analysis is presented in the Supplementary Fig. S6f. **e** Plot showing the number of intermolecular hydrogen bonds (H bonds) formed/broken between the ECL2 region (residues 174–196) of C5aR (WT) and PMX53 over the course of the pulling simulation. **f** Evolution of key intramolecular interaction R6PMX53–W5PMX53 cation-π interaction (green) in PMX53 over the course of the pulling simulation. **g** Position and conformation of PMX53 at $t = 0$ ps during pulling simulation (pull force = $7.62 \times 10^{-5}$ pN) where R6PMX53 stably and directly interacts with D282C5aR as compared to the conformation observed in the starting crystal structure conformation. In this conformation, PMX53 forms extensive H bond interactions (shown as black lines) with the residues of C5aR (WT), especially with residues of ECL2. **h** Position and conformation of PMX53 at $t = 325$ ps during pulling simulation (pull force = 2386.14 pN) where key non-covalent interactions between PMX53 and C5aR (WT) begin to break and R6PMX53 and W5PMX53 are being pulled away from Y258C5aR and D282C5aR. A number of HBonds between PMX53 and ECL2 also as broken or are in the process of being broken under the influence of the applied force. **i** Position and conformation of PMX53 at $t = 425$ ps during pulling simulation (pull force = 879.08 pN) where the PMX53 molecule has been pulled further away with the R6PMX53–D282C5aR salt-bridge and the R6PMX53–Y258C5aR cation-π interaction being completely broken. **j** Position and conformation of PMX53 at $t = 600$ ps during pulling simulation (pull force = 29.48 pN) where the ligand is completely unbound from the receptor.

intermolecular hydrogen bonds, especially with ECL2 (Fig. S5a, b). Key intermolecular interactions between C5aR and PMX53 present in the initial crystal structure such as the D282-R6PMX53 salt-bridge remained stable throughout the entire 300 ns (Fig. S5c). The Y258–R6PMX53 cation-π interaction was broken (R6PMX53 CZ and Y258 ring–centroid distance >6.0 Å) halfway through the simulation but the two residues remained close to each other (Fig. 4g). Disruption of the cation-π interaction allowed R6PMX53 to interact with D282 in a head-on manner (Fig. 4g and Fig. S5c). W5PMX53 and R6PMX53 saddled Y258 but did not interact directly during the production run (Fig. 4g and Fig. S5c).

**PMX53 dissociates from C5aR in two critical steps**. We employed SMD or center-of-mass (COM) pulling simulations on the final configuration of the 300 ns equilibrium simulation to gain an atomistic insight into the molecular events that occur during the dissociation of PMX53 from the C5aR binding pocket. SMD has been successfully employed for studying biological phenomenon such as stability of α-amyloid protofibrils[21], substrate translocation by membrane transporters[22], and interaction of GPCR ligands with their cognate receptors[23]. Akin to AFM experiments, in the pulling simulations, the bound PMX53 molecule was pulled away from the C5aR binding pocket by

applying an external force along the $z$-axis (Fig. 4a, b and Supplementary Movie 1). The force vs. time profile of the pulling simulation is presented in Fig. 4b. The application of force on the PMX53 molecule led to a gradual build-up of force until a critical point was reached that was sufficient to break the key intermolecular interactions to allow the dissociation of the bound molecule. The plot showed two such critical points, a minor drop in force around $t = 308$ ps and a major drop in force around $t = 425$ ps. After the major drop, the PMX53 molecule was mostly unbound. We analyzed the evolution of various intermolecular non-covalent interactions between C5aR and PMX53 during the pulling simulation (Fig. 4c, e, f). The analysis revealed that shortly after $t = 308$ ps time-point numerous hydrogen bonds, almost half of which were formed between the ligand and the residues of the C5aR ECL2 region, were broken, resulting in a brief drop in force (Fig. 4e, h). Further, the critical D282-R6$_{PMX53}$ salt-bridge and the Y258-R6$_{PMX53}$ cation-π were completely broken around $t = 425$ ps time-point when the pulling force was maximal (Fig. 4i, f). Following the breakage of these critical interactions, the PMX53 molecule adopted a more compact conformation facilitated by the formation of an intramolecular R6$_{PMX53}$–W5$_{PMX53}$ cation-π interaction (Fig. 4f, j).

We also performed umbrella sampling simulations on the configurations generated from the non-equilibrium SMD trajectories to calculate the free energy profile of the PMX53 binding/dissociation events. The weighted histogram analysis method (WHAM) was employed to obtain the potential of mean force (PMF) curve from which the $\Delta G$ of PMX53 binding was deduced (Fig. 4d). Bootstrap analysis was used to estimate the statistical errors, and the average PMF profile along with the corresponding standard deviation values are plotted in Fig. S6f. On the basis of the PMF profile, we obtained a $\Delta G_{bu}$ value of $-13.8$ kcal mol$^{-1}$.

**MD simulations identify key residues involved in antagonist binding to C5aR.** Based on the MD simulations, we tried to determine the key receptor residues involved in ligand binding. As PMX53 structurally mimics the structure of the C-terminal segment of the cognate C5a ligand[2,24,25], we can hypothesize that they could interact through similar residues. MD and SMD simulations pointed out two critical interactions established in the effector site, involving C5aR residues R175, D282, and Y258. Indeed, in the crystal structure of C5aR with PMX53[8], both R175 and D282 form direct polar interactions with PMX53, and Y258 forms cation-π interactions with an arginine residue of PMX53. We designed two C5aR mutants, C5aR$^{D282A}$ and C5aR$^{R175V/Y258V}$, and performed GTPγS binding assays with C5aR$^{WT}$ and the two mutants. We observed a strong reduction in the G$_i$-protein activation induced by two C5aR mutants in response to increased concentration of C5a (Fig. S7), confirming the crucial role played by these residues in the modulation of C5aR functional state.

**C5aR$^{R175/Y258}$ dictates the binding kinetics.** Next, we used FD-based AFM to evaluate the kinetics and thermodynamics implications of C5aR mutations within the effector site by probing the interaction using PMX53 functionalized AFM tips (Fig. 3i, j). Thermodynamic analysis using the FNdY model only revealed a slight reduction of the $\Delta G_{bu}$ from $-13.7 \pm 4.9$ kcal mol$^{-1}$ for the C5aR$^{WT}$ to $-13.3 \pm 1.1$ kcal mol$^{-1}$ for the C5aR$^{R175V/Y258V}$ double mutant (Fig. 3i). MD and SMD simulations using the PMX53–C5aR$^{R175V/Y258V}$ double mutant system were performed following the same protocol used for the PMX53–C5aR$^{WT}$ complex (Figs. S5, 6). These in silico experiments confirmed the minor reduction in $\Delta G_{bu}$ observed by AFM and provided more insights into the mechanism governing this interaction. The most striking observation was the reduction in the number of hydrogen

bonds formed between C5aR$^{R175V/Y258V}$ and PMX53, particularly involving residues from ECL2. The R175V mutation causes a 66% reduction in the number of hydrogen bonds formed between ECL2 and PMX53 ($1.57 \pm 0.91$) as compared to the wild type ($4.62 \pm 0.93$; Fig. S5b). The R6$_{PMX53}$–D282 salt-bridge remained stable throughout the 300 ns production run, whereas the Y258V mutation caused a change in the stability of the W5$_{PMX53}$ orientation (Fig. S5c, d). When PMX53 was pulled away from C5aR$^{R175V/Y258V}$ using a similar SMD protocol, we observed a marked drop in the force required for the ligand to dissociate (Fig. S6b). The inter- and intramolecular non-covalent interactions behave in a similar fashion as the wild type, but break much earlier (Fig. S6c–e). The hydrogen bonds between ECL2 and PMX53 broke much earlier around $t = 200$ ps (Fig. S6e) as compared to the wild type ($t = 308$ ps), while the R6$_{PMX53}$–D282 salt-bridge breaks shortly thereafter, but earlier than the wild type (Fig. S6d). The PMF profile for C5aR$^{R175V/Y258V}$ shows a significant drop (~43%) in the height of the energy barrier crossed during PMX53 dissociation ($-12.2$ kcal mol$^{-1}$ for double-mutant vs. $-21.5$ kcal mol$^{-1}$ for WT), although resulting in a slight reduction (~12%) in $\Delta G_{bu}$ ($-12.2$ kcal mol$^{-1}$ for C5aR$^{R175V/Y258V}$ vs. $-13.8$ kcal mol$^{-1}$ for C5Ar$^{WT}$) (Fig. S6f). Results from our MD and SMD studies are in good agreement with experimental data obtained by AFM where we observed a slight decrease in the $\Delta G_{bu}$ (~3%) but a much important reduction in residence time (~40%) that can be directly linked with the reduction of the height of the energy barrier crossed during PMX53 dissociation. Altogether, these results suggest that the R175 and Y258 residues play a key role in the kinetics of the interaction, although the thermodynamics seems to be dictated by other residues.

**C5aR$^{D282}$ is critical for binding thermodynamics.** We also studied the PMX53 binding to the C5aR$^{D282A}$ mutant by FD-based AFM (Fig. 3j). The analysis of the DFS plot with the FNdY model revealed a strong reduction of the free energy (~43%) leading to a $\Delta G_{bu}$ of $-7.7 \pm 1.9$ kcal mol$^{-1}$, while the residence time remains unchanged (0.1 ms). MD and SMD simulations were attempted on this mutant, with no success in obtaining convincing results for the umbrella sampling simulations due to largely reduced PMX53–C5aR$^{D282A}$ interactions. Yet, both experimental and simulation experiments suggest a strong reduction of the interactions due to the single point mutation in the receptor, thus underlying the important role of D282 in the stabilization of the PMX53 into the binding pocket. These results are also in good agreement with the functional assays revealing that the D282 residue strongly modulates C5aR functional state of the receptor (Fig. S7).

**High-affinity C5a binding results from positive allosteric interactions between the effector and binding sites.** Having shown that we can monitor the binding to C5aR effector site, we next wanted to decipher the binding free energy landscape of the full C5a–C5aR complex (Fig. 5). AFM tips were functionalized with the C5a ligand and probed against C5aR$^{WT}$ using FV- and FD-based AFM (Fig. 5a). The binding properties of the C5a–C5aR complex determined by the FNdY model are characterized by an equilibrium force $F_{eq}$ of $46 \pm 7$ pN, a binding equilibrium free energy $\Delta G_{bu}$ of $-13.6 \pm 4.1$ kcal mol$^{-1}$, and a dissociation constant in the high-affinity range, $K_d$ of ≈4.5 nM (Fig. 5a). Despite being a large molecule compared to the PMX53 antagonist, C5a shows a similar affinity for C5aR. As previous studies suggest a two-site binding mechanism, we scrutinized the role of the binding site and effector site at the submolecular level using receptor mutants and blocking experiments.

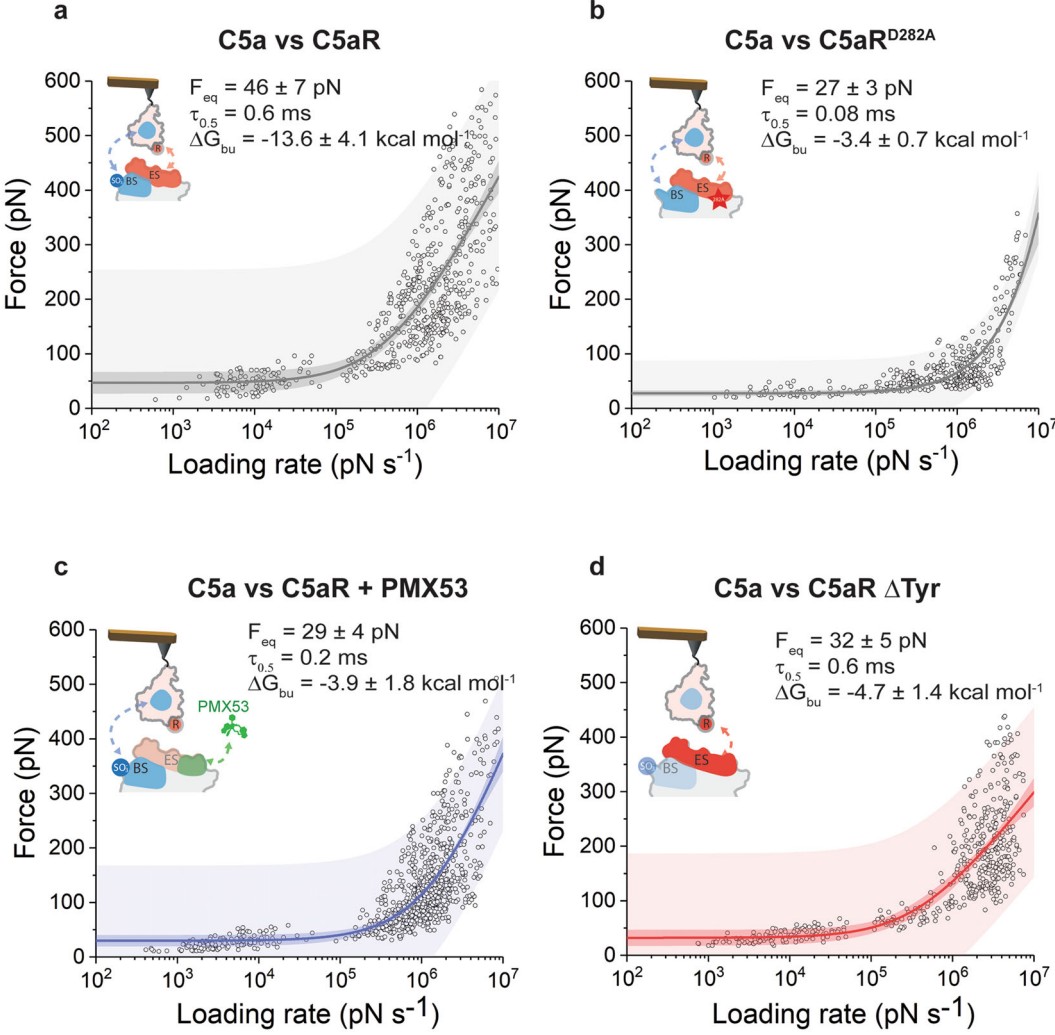

**Fig. 5 Submolecular probing of the kinetic and thermodynamic parameters underlying C5a ligand binding to C5aR.** DFS plots showing the loading rate-dependent interaction forces of the C5a ligand probed against wild-type C5aR (**a**) and C5aR$^{D282A}$ (**b**). **c**, **d** Probing the kinetic and thermodynamic parameters underlying C5a ligand binding to the orthosteric binding sites. The binding site (**c**) was probed using C5aR complexed in presence of the PMX53 antagonist. To access the effector site (**d**), C5a was probed with C5aR missing a Tyr residue at the N-terminal end. Fitting the data using the Friddle–Noy–de Yoreo model (thin lines) provides $F_{eq}$, $\Delta G_{bu}$, and residence time ($\tau_{0.5}$) values with errors representing the s.e.m. Each circle represents one measurement. Darker shaded areas represent 99% confidence intervals, and lighter shaded areas represent 99% of prediction intervals. For each condition, data are representative of at least three independent experiments.

First, we observed that the mutation of the D282 residue in the C5aR has a strong destabilizing effect, with a four-fold drop of the $\Delta G_{bu}$ ($-3.4 \pm 0.7$ kcal mol$^{-1}$), towards the low-affinity regime (Fig. 5b). The residence time also sees an important decrease as a result of the D282 mutation, from 0.6 ms for the wild-type C5aR to 0.08 ms for C5aR$^{D282A}$. This result also confirms previous findings, which point the effector site as key player in the ligand-binding and activation process of C5aR.

Next, to address specifically the role of the binding site, we used a C5a derivatized AFM tip and incubated the C5aR$^{WT}$ with PMX53, to block the interactions at the effector site (Fig. 5c). Fitting the DFS plot with the FNdY model gave an equilibrium force $F_{eq}$ of $29 \pm 4$ pN, revealing that the inner core of the C5a binds to C5aR binding site with free-energy values ($\Delta G_{bu}$) of $-3.9 \pm 1.8$ kcal mol$^{-1}$, which corresponds to very high dissociation constant $K_d$ of ≈0.8 M. We also looked into the kinetics of the C5a-binding site interaction and quantified the complex stability in terms of residence time. A $\tau_{0.5}$ value of 0.2 ms was

obtained, which is three times lower than the one observed for C5a–C5aR (Fig. 5a).

Finally, to obtain the full picture of the binding mechanism between C5a and its receptor, we also investigated the binding interaction between the C-terminal segment of C5a and the effector site. To achieve this, we abolished the interaction that the rigid core of C5a establishes with sulfonated residues at the binding site, so the C5aRΔTyr mutant with mutation sites Y11F and Y14F was generated[6]. The lack of sulfonation on the C5aRΔTyr mutant was validated by western blot using anti-sulfated tyrosine antibodies (Fig. S1). The interaction between the C5a ligand and C5aRΔTyr was measured and the dependence of the rupture force with the loading rate was plotted in the DFS graph in Fig. 5d. We extracted an equilibrium force $F_{eq}$ of $32 \pm 5$ pN, a binding equilibrium free energy $\Delta G_{bu}$ of $-4.7 \pm 1.4$ kcal mol$^{-1}$, and a receptor–ligand half-residence time, $\tau_{0.5}$ of 0.6 ms. The calculated $\Delta G_{bu}$ corresponds to a dissociation constant $K_d$ of ≈20 mM, which points to a surprisingly low-affinity. Interestingly, despite

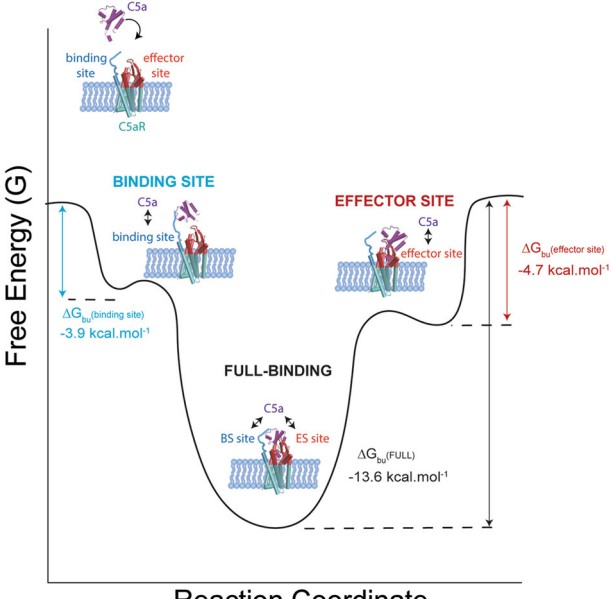

**Fig. 6 Illustration of the free-energy binding landscape of C5a binding to C5aR.** $\Delta G_{bu}$ gives the free-energy difference between the ligand-bound and unbound states and is indicated for each binding site (binding site, effector site, and binding site + effector site) by vertical arrows. A positive allosteric interaction is measured when both binding sites (binding site and effector site) are occupied, as revealed by a significantly higher $\Delta G_{bu}$ for the full binding of the C5a.

the reduced affinity, the effector site seems to interact with C5a with similar kinetic parameters as the full receptor.

Taken together, these results point towards a cooperativity mechanism, through a positive allosteric effect, between the two binding sites (Fig. 6). By taking a closer look at the thermodynamic parameters of the interactions, we observe that the full-ligand binding free-energy ($\Delta G_{bu}(C5a) \sim -13.6 \pm 4.1$ kcal mol$^{-1}$) is significantly higher than the sum of the binding free-energy of both sites measured individually ($\Delta G_{bu\ (binding\ site+effector\ site)}) \sim -8.6 \pm 2.3$ kcal mol$^{-1}$). In line with our results, intrinsic binding energies have been previously described for intramolecular binding events involving two separate interacting regions of the same molecule[26]. Multi-step binding processes could be stabilized through a decrease in translational and rotational entropy when the first interaction forms. This supports a positive allosteric interaction between the two orthosteric binding sites, establishing the full interaction with the C5a. This is consistent with previous studies showing that mutations of Y11F and Y14F could almost abolish the signaling of C5aR induced by C5a[6]. Our AFM experiments further suggest that the interaction between C5a and C5aR-D282 plays a pivotal role in this cooperative mechanism. Indeed, the single point mutation into the C5aR (D282A) is sufficient to completely abolish this high-affinity interaction state.

## Discussion
GPCRs represent the largest human membrane protein family, having overall more than 800 members, and constitute a "control panel" of the cell[27]. As predominant actors in cells, GPCRs are intensively studied as drug targets, where, in particular, C5aR has long been suggested as a new promising anti-inflammatory target. Intensive research on C5aR has led to the design of several antagonists including the peptide antagonist PMX53 and several non-peptide antagonists such as NDT9513727 and avacopan. PMX53 is a potent orthosteric antagonist with insurmountable action[19]. However, the peptidic nature has limited its clinical

development[3]. Among the current available non-peptide antagonists, only avacopan showed sufficient therapeutic efficacy to advance into late-stage clinical trials[28,29]. Recent structural studies on C5aR revealed that the non-peptide antagonists (including avacopan and NDT9513727) are actually allosteric modulators with highly reversible action[8,30]. Further development of orthosteric non-peptide antagonists could be preferred as they may exhibit an insurmountable action similar to PMX53. Corroborated with previous studies revealing the structural basis for the action of PMX53[8], our kinetic and thermodynamic insights of PMX53 binding to the receptor effector site confirmed by MD and SMD simulations, shed more light into the activation mechanism of C5aR and the amino acid residues involved, which could be useful for future drug discovery studies.

Atomic-resolution structures are now available for more than 50 different GPCRs and over 250 of their complexes with different ligands[31]. Crystal structures of C5aR in complex with NDT9513727, PMX53, and avacopan have recently been reported[8,30]. However, despite the dramatic progress during the last decade in deciphering the structural insights of C5aR activation mechanism, none of those recent structural studies have been performed with the C5a ligand. In addition, the function of GPCRs depends critically on their ability to change shape, transitioning among distinct conformations, while crystal structures only depict discrete snapshots of a dynamic process. Although for some GPCRs, several small drug candidates have been developed using solely structure-based drug design methodologies[32], a full understanding of the dynamic properties of GPCRs is preferred and probably essential for future drug development, especially for those with large peptide or protein ligands. Here, we introduced an FD-based AFM approach and a new experimental strategy to extract the kinetic and thermodynamic parameters governing large-ligand binding to multiple orthosteric binding sites of receptors in physiologically relevant conditions. We also used MD and SMD simulations as a powerful complementary method to our experimental approach, allowing us to gain new insights into the binding pocket structure and the important residues involved in the specific recognition of ligands.

Our study addresses the complex binding process of a large ligand to a GPCR. C5a, a large 74-amino acid glycoprotein, binds to C5aR through two distinct and physically separated binding sites, namely the effector site and the binding site[5]. While the existence of the two-site binding motif has been previously reported[5], the energetic contribution of these sites to the overall ligand affinity, as well as any positive allostery between them, remained unclear.

We turned to in silico experiments and FD-based AFM to answer this question. We used the unique capabilities of AFM over some of the more traditional approaches, for instance isothermal titration calorimetry, that allows to resolve not only the thermodynamic, but also the kinetic parameters of the binding process at the single-molecule level in a simultaneous fashion. Our method enabled, for the first time, to probe multiple ligand-binding sites at the sub-site level in order to study their respective contribution to the overall binding. We were able to capture the "cryptic" binding pockets of C5a into C5aR and to reconstruct the binding free-energy landscape for this complex binding mechanism. We demonstrated that both orthosteric ligand-binding sites interact with the ligand with a low-affinity when working independently and that ligand binding in its high-affinity state involves the concerted action of both sites. Together with steered MD simulations that recapitulate AFM results, our observations point toward an allosteric model in which C5a binding to one site does not enhance the intrinsic affinity of the receptor for the other, but rather acts as a kinetic trap, effectively boosting local ligand concentration and increase binding (Fig. 6).

The importance of the D282 at the extracellular face of TM7 was also put in evidence as a key factor in the thermodynamics of the ligand-binding process.

Notably, C5a also binds and activates a second receptor, C5aR2, albeit without any detectable G-protein coupling but with robust β arrestins (βarr) recruitment[33]. Three N-terminal tyrosine residues of C5aR2 may also be sulfated, but mutations of these residues did not significantly change C5a binding to C5aR2[34], suggesting against the two-site binding mechanism for C5aR2. Our mechanistic insights into the C5a binding process to C5aR could offer a new framework to a better understanding or the intriguing functional divergence between the two C5a receptors and could potentially allow delineating the link between βarr conformational signatures and downstream functional outcomes[35]. Moreover, since a similar two-step two-site binding model has also been proposed for chemokine–receptor interactions[36,37], our approach could also be useful to further study phenomena such as allosteric receptor interactions and ligand-biased receptor activation in this context as well.

This study presents a detailed kinetic and thermodynamic analysis of the mechanism by which C5a anaphylatoxin interacts with its target C5aR, a G-protein-coupled receptor whose signaling plays a critical role in a variety of immunological responses. Using AFM, we probed ligand binding at the sub-site level in physiologically relevant conditions and extracted the kinetic and thermodynamic parameters of the interaction. Taken individually, the two binding sites at the receptor side interact with the C5a ligand at low-affinity. Our results evidence for the first time a cooperative mechanism between the two binding sites, establishing a high-affinity interaction with the C5a ligand, which in turn enables C5aR activation. By solving this longstanding paradigm underlying the binding of large ligand to GPCRs, we open new avenues for the development of new pharmacotherapies for inflammatory diseases. We envision that this better understanding of the dynamic binding of C5a to C5aR in physiologically relevant conditions will open new avenues in the rational design of finely tuned drugs. Ultimately, this approach will serve as a valuable tool to further develop and test agonists and antagonists to other GPCRs with macromolecular ligands.

## Methods

**wtC5aR Expression, purification, and western blot**. The wild type C5aR and mutant were expressed in mammalian HEK-293S GnTI⁻ cells (ATCC) using the BacMam method[38]. All constructs were cloned into a vector engineered from pFastBac (Invitrogen) by introducing a CMV promoter[38]. All proteins were expressed with a C-terminal His$_8$-tag and an N-terminal Flag tag. Baculovirus was generated by the Bac-to-Bac method (Invitrogen). The mammalian HEK-293S GnTI⁻ cells were cultured in suspension at 37°C and under 5% CO$_2$. The cells were infected at a density of $4 \times 10^6$ ml⁻¹ with baculovirus and then harvested after 24 h.

To purify the protein, infected cells were lysed in buffer containing 10 mM Tris pH 7.5, 150 μg ml⁻¹ benzamidine, 0.2 μg ml⁻¹ leupeptin, and 2 mg·ml⁻¹ iodoacetamide. The cell membrane was collected by centrifugation at 24,000g for 40 min at 4 °C and then solubilized in buffer containing 20 mM HEPES pH 7.5, 750 mM NaCl, 1% dodecyl maltoside (DDM), 0.2% cholesterol hemisuccinate (CHS), 0.2% sodium cholate, 20% glycerol, 150 μg ml⁻¹ benzamidine, 0.2 μg ml⁻¹ leupeptin, 2 mg ml⁻¹ iodoacetamide, and 5 U l⁻¹ Salt Active Nuclease (Sigma) for 1 h at 4 °C. The supernatant was collected after centrifugation at 24,000g for 40 min, and incubated with Ni-NTA agarose resin (Clontech) in batch for overnight at 4 °C. The resin was washed three times in batch with buffer comprising of 20 mM HEPES pH 7.5, 500 mM NaCl, 0.1% DDM, 0.02% CHS, 150 μg ml⁻¹ benzamidine, 0.2 μg ml⁻¹ leupeptin, and 20 mM imidazole, and then transferred to a gravity column. After extensive washing, the protein was eluted in wash buffer with 400 mM imidazole and 2 mM CaCl$_2$. The eluted protein was loaded onto anti-Flag M1 antibody affinity resin. After washing with buffer containing 20 mM HEPES pH 7.5, 100 mM NaCl, 0.1% DDM, 0.02% CHS, and 2 mM CaCl$_2$, the protein was eluted with buffer containing 20 mM HEPES, pH 7.5, 100 mM NaCl, 0.1% DDM, 0.02% CHS, 200 μg ml⁻¹ Flag peptide, and 5 mM EDTA. The protein was further purified by size exclusion chromatography with buffer containing 20 mM HEPES pH 7.5, 100 mM NaCl, 0.05% DDM, and 0.01% CHS.

Mouse anti-Flag M1 antibody and mouse anti-Sulfotyrosine antibody (Sigma) were used to detect the purified wild type C5aR and C5aRΔTyr with Y11F and F14F mutations, respectively, in the western blotting assays.

**C5aR mutants expression, purification, and 35S-GTPγS binding assay**. Mutant variants (D282A, D282N, and R175V/Y258V) were generated based on the wtC5aR construct and fully sequenced. Mutant variants were expressed following the same method as for wtC5aR except for some modifications. HEK-293S cells expressing each mutant were pelleted by centrifugation and resuspended in 20 ml buffer containing 20 mM HEPES pH 7.5, 100 mM NaCl, 0.2 μg ml⁻¹ leupeptin, and 150 μg ml⁻¹ benzamidine. After 20 min incubation at 25 °C, 20 ml 2X solubilization buffer containing 20 mM HEPES pH 7.5, 100 mM NaCl, 1% dodecyl-maltoside (DDM), 0.2% cholesterol hemisuccinate (CHS), 20% glycerol, 0.2 μg ml⁻¹ leupeptin, 150 μg ml⁻¹ benzamidine, and 5 U Salt Active Nuclease (Sigma) was added. Cell membranes were solubilized for 1.5 h at 4 °C. The supernatant was collected by centrifugation at 24,000g for 30 min at 4 °C, and then incubated with anti-Flag M2 antibody affinity resin (Sigma) for 1.5 h at 4 °C. After washing the resin with buffer containing 20 mM HEPES pH 7.5, 100 mM NaCl, 0.1% DDM, 0.02% CHS, 0.2 μg ml⁻¹ leupeptin, and 150 μg ml⁻¹ benzamidine, the protein was eluted from M2 resin using the buffer containing 20 mM HEPES pH 7.5, 100 mM NaCl, 0.1% DDM, 0.02% CHS, and 200 μg ml⁻¹ Flag peptide (GL Biochem). The protein was further purified by size exclusion chromatography with the same buffer as for wtC5aR.

For the ³⁵S-GTPγS binding assays, the membrane of HEK293S GnTI⁻ cells expressing wtC5aR (~200 μg ml⁻¹) or mutant variants was incubated with 200 nM purified G$_i$ protein for 30 min on ice in buffer containing 20 mM HEPES pH 7.5, 100 mM NaCl, 5 mM MgCl$_2$, 3 μg ml⁻¹ BSA, 0.1 μM TCEP, and 5 μM GDP to get the receptor and G$_i$ complex. Next, 25 μL aliquots of the pre-formed complex were mixed with 225 μL reaction buffer containing 20 mM HEPES, pH 7.5, 100 mM NaCl, 5 mM MgCl$_2$, 3 μg ml⁻¹ BSA, 0.1 μM TCEP, 1 μM GDP, 35 pM ³⁵S-GTPγS (Perkin Elmer), and C5a (R&D Systems). After additional 15 min incubation at 25 °C, the reaction was terminated by adding 5 ml of cold wash buffer containing 20 mM HEPES pH 7.5, 100 mM NaCl, and 5 mM MgCl$_2$, and filtering through glass fiber filters (Millipore Sigma). After washing the filters twice with 5 ml cold wash buffer, the filters were incubated with 5 ml of CytoScint liquid scintillation cocktail (MP Biomedicals). The radiation of bound ³⁵S-GTPγS was measured on a Beckman LS6500 scintillation counter to determine the binding of ³⁵S-GTPγS to G$_i$ induced by C5aR activation. The data analysis was performed using GraphPad Prism 6 (GraphPad Software). Results are shown as mean ± s.d. from three independent experiments.

**C5aR liposomes preparation**. C5aR liposomes were prepared according to previously published method[10]. The empty liposomes were prepared from a mix of DOPC (1,2-Dioleoyl-sn-glycero-3-phosphocholine) (Avanti lipids) and CHS (Sigma). DOPC and CHS were dissolved in chloroform at a 10:1 (w:w) ratio, then mixed and dried. The well-mixed DOPC/CHS was re-suspended and dissolved in buffer containing 20 mM HEPES pH 7.5, 100 mM NaCl, and 1% octylglucoside (OG) under sonication on ice. Aliquots of dissolved DOPC/CHS lipids were flash-frozen in liquid nitrogen and stored at −80 °C.

To reconstitute C5aR in liposomes, protein and lipids were mixed at a 10 μM:1 mM final ratio, and incubated on ice for 2 h. The detergent was removed by biobeads (Bio-rad) and extensive dialysis.

**C5aR preparation for AFM measurements**. The reconstituted C5aR sample solution (either wt-C5aR or mutants) was 20-fold diluted in fusion buffer solution (20 mM HEPES, 300 mM NaCl, and 25 mM MgCl$_2$) and adsorbed on freshly cleaved mica for 15 min. After rinsing with imaging buffer (20 mM HEPES and 300 mM NaCl) the sample was transferred to the AFM.

**Functionalization of AFM tips**. Rectangular Si$_3$N$_4$ AFM cantilevers with silicon tips (BioLever mini, Bruker) were first cleaned with chloroform for 10 min, rinsed with ethanol, N$_2$ dried, and then cleaned for 15 min in an ultraviolet radiation and ozone cleaner (UV-O, Jetlight, CA, USA). For the aminofunctionalization, the cantilevers were immersed in an ethanolamine solution (3.3 g ethanolamine in 6.6 ml DMSO) overnight and then rinsed in DMSO (3 × 1 min) and ethanol (3 × 1 min), followed by N$_2$ drying[39]. This was followed by the N-hydroxysuccinimide (NHS)-PEG$_{27}$-acetal linker attachment. A 1 mg portion of the NHS-PEG$_{27}$-acetal linker (JKU, Linz, Austria) was diluted in 0.5 ml chloroform with 30 μl triethylamine and the cantilevers were immersed in this solution for 2 h. After three rinsing steps of 10 min in chloroform and N$_2$ drying, the cantilevers were immersed in a 1% citric acid solution for 10 min, rinsed with pure water (3 × 5 min), and dried with N$_2$ once more. tris-NTA-derivatized AFM cantilevers were obtained by pipetting 100 μl of a 100 μM tris-N-nitrilotriacetic amine trifluoroacetate (Toronto Research Chemicals, Canada) (tris-NTA) solution onto the cantilevers, followed by the addition of 2 μl of a freshly prepared 1 M NaCNBH$_3$ solution. The cantilevers were incubated for 1 h, then 5 μl of a 1 M ethanolamine solution pH 8.0 was added for 10 min to quench the reaction. tris-NTA cantilevers were further used to obtain C5a-derivatized tips. For this purpose, 100 μl of a 1 μM C5a protein solution was premixed with 5 μl NiCl$_2$ 5 mM and the mixture was pipetted onto the tris-NTA

cantilevers. After 2 h of incubation time, the cantilevers were washed in HEPES buffer 3 × 5 min.

To functionalize AFM cantilevers with the PMX53 antagonist (Ace-Phe-[{Orn}-Pro-{D-Cha}-Trp-Arg]), aminofunctionalized cantilevers were immersed for 2 h in a solution prepared by mixing 1 mg of NHS-PEG$_{27}$-maleimide[39] (JKU, Linz, Austria) dissolved in 0.5 ml of chloroform with 30 μl of triethylamine, then washed with chloroform and dried with $N_2$. The cystein bearing peptide Cys-Gly$_3$-Phe-[{Orn}Pro-{D-Cha}-Trp-Arg] (PMX53-Gly$_3$-Cys) was obtained from GL Biochem (Shanghai). A 100 μl solution of Cys-Gly$_3$-PMX53 1 mM was premixed with 2 μl of EDTA (100 mM, pH 7.5), 5 μl of HEPES (1 M, pH 7.5), 2 μl of TCEP hydrochloride (100 mM), and 2 μl of HEPES (1 M, pH 9.6), then pipetted over the AFM cantilevers. After 3 h of reaction, cantilevers were washed with PBS 3 × 5 min.

**FD- and FV-based AFM**. AFM experiments were performed with a Multimode 8 AFM equipped with a Nanoscope V controller (Bruker, Santa Barbara, CA, USA) operated in "PeakForce Tapping QNM mode". All measurements were carried out in imaging buffer at room temperature (≈24 °C). For the high-resolution characterization of C5aR topographical features, triangular $Si_3N_4$ cantilevers (Scanasyst-Fluid+, Bruker) with a sharpened tetrahedral silicon tip of ≈2 nm radius, nominal spring constants of 0.35 N m$^{-1}$, and resonance frequency in liquid of ≈75 kHz were used. Multiparametric FD-based AFM measurements with derivatized tips were carried out using BioLever mini cantilevers (Bruker, Santa Barbara) having nominal spring constants of 0.1 N m$^{-1}$ and resonance frequency in liquid of ≈25 kHz. The spring constant was calibrated at the end of each experiment for all cantilevers used in this study using the thermal noise method[40].

In FD-based AFM measurements, the AFM cantilever is oscillated in a sinusoidal manner well below its resonance frequency, while the sample surface is contoured pixel-by-pixel. For each approach–retraction cycle of the oscillating cantilever, a force–distance curve is recorded. Multiparametric FD-based AFM height, Young's modulus, and adhesion maps are then obtained from a pixel-by-pixel reconstruction of the acquired data. Overview FD-based AFM maps were acquired by scanning the sample at 1 Hz and a resolution of 512 × 512 pixels, using a force setpoint of ≈150 pN, a 2 kHz oscillation frequency, and a peak-to-peak oscillation amplitude of 100 nm. Adhesion maps were recorded using a scan rate of 0.2 Hz and 256 × 256 pixels. The functionalized AFM cantilever was oscillated at 0.25 kHz with peak-to-peak oscillation amplitudes of 100 nm.

To vertically oscillate the AFM tip at 1–10 Hz, FV-based AFM was conducted in the ramp mode with a force setpoint of 200 pN, an approach velocity of 1 μm s$^{-1}$, retraction velocities of 0.5–2 μm s$^{-1}$, a ramp size of 150 nm, and no surface delay.

To increase the statistics and to show reproducibility of the experiments, for each experimental condition, we prepared at least five different functionalized AFM tips and imaged at least three independent C5aR sample preparations ($n = 3$–10).

**Control experiments**. Several control experiments were designed to ensure that the measured interactions were indeed specific and the functionalization of the AFM tip successful. Adhesion maps of C5aR reconstituted samples were imaged with unmodified or ethanolamine-coated AFM tips (Fig. S3a–d). We also tested C5a ligand binding before and after injection of free C5a on the sample surface (Fig. S3e, f). In another approach, tris-NTA binding to C-terminal of C5aR was tested in the presence of 10 mM EDTA (Fig. S3g, h).

**Data analysis**. Raw images were analyzed using the Gwyddion 2.5 free software. Force–distance curves were analyzed using the Nanoscope Analysis 1.80 Software (Bruker). Individual force–distance curves corresponding to specific adhesion events were extracted and further analyzed using the OriginLab software. Adhesion forces were calculated as the minimum force in the retraction segment of the force–distance curve and the loading rate was measured as the slope of the force vs. time curve just before rupture. The noise level was calculated by doing a linear fit of the retraction part of the force–distance curve and calculating the standard deviation. We obtained noise values between 10 and 15 pN and set a threshold for the specific unbinding events above 25 pN. Dynamic force spectroscopy (DFS) graphs were obtained by plotting the loading-rate dependence of the adhesion force and a nonlinear iterative fitting algorithm (Levenberg-Marquardt) was used with the FNdY model to extract kinetic and thermodynamic parameters of the interactions. The fits were plotted along with the 99% confidence intervals and 99% prediction intervals. Each DFS plot includes between 200 and 700 data points and each data point represents a binding event between the functionalized AFM tip and C5aR particles on the sample.

**Molecular dynamics simulation system setup**. The dual antagonist-bound C5aR structure complexed with PMX53 and avacopan in the orthosteric and allosteric sites, respectively, solved by Liu et al. [PDB ID: 6C1R] was used for setup of the simulation systems. All atoms other than those of C5aR and PMX53 (avacopan, solvent, lipids, etc.) were removed along with the engineered N-terminal cytochrome $b_{262}$ RIL (BRIL). All non-terminal missing regions (234–236, 308–312) were modeled using MODELLER v9.13 via the Model Loops/ Refine Structure module available in UCSF Chimera[41]. A total of 500 structures with the missing loop regions were modeled and the one with the best zDOPE score was selected for preparation of the system for molecular dynamics (MD) simulations.

For the PMX53–C5aR double-mutant system, the R175V and Y258V mutations were introduced into the WT-C5aR–PMX53 system using the Rotamers module available in USCF Chimera. The N- and C-termini of C5aR were acetylated and amidated, respectively. The C5aR–PMX53 complex was then embedded in a lipid bilayer comprising 164 POPC (1-palmitoyl-2-oleoyl-sn-glycerol-3-phosphocholine) molecules (82 each on the upper and lower layers) using the CHARMM-GUI server. The receptor–antagonist–lipid system was then solvated with 27,000 TIP3P water molecules, and NaCl at a concentration of 150 mM was added. The final dimensions of the system were ~79.1 Å × 79.1 Å × 170 Å comprising ~108,000 atoms. CHARMM36 force field[42] parameters were used to model protein, lipids, ions, and water molecules. For PMX53, force field parameters were assigned by analogy using CHARMM general force field (CGenFF) via the ParamChem server[43].

GROMACS v5.1.2[44] was used for performing all the simulations. Short-range non-bonded interactions were calculated with a 1.2 nm cut-off, and the particle mesh Ewald (PME) algorithm[45] was employed for calculation of long-range electrostatics. LINCS[46] algorithm was used to constraint all H-atom containing bonds. The system was first energy minimized using steepest decent algorithm. Subsequently, the system was equilibrated in a stepwise manner, first in an NVT ensemble (three steps, 50 ps each with 1 fs time step) maintained at 310 K by Berendsen coupling. The system was then equilibrated in an NPT ensemble (three steps, 100 ps each with 2 fs time step) maintained at 310 K and 1.0 bar using Berendsen coupling. The harmonic position restraints applied to the heavy atoms of C5aR, PMX53, and POPC were reduced gradually at each of the six equilibration steps to ensure thorough equilibration of the system. Following equilibration, all restraints were removed and production runs were carried out by maintaining the temperature (310 K) and pressure (1.0 bar) with the help of Nosé-Hoover thermostat and Parrinello-Rahman barostat, respectively. Pressure coupling was carried out semi-isotropically for NPT equilibration and production runs. Finally, a production run of 300 ns was carried out.

**Center-of-mass pulling and umbrella sampling simulations**. The resultant configuration of the 300 ns production run was used for performing the COM pulling simulations. The final configuration was equilibrated for 100 ps in an NPT ensemble. Subsequently, with positional restraints placed only on the C5aR molecule in the z-direction, the bound PMX53 cyclic peptide was pulled away from C5aR binding pocket. The pulling simulation was carried out over 1 ns along the z-axis with a pull rate of 0.005 nm ps$^{-1}$ and spring constant of 1000 kJ mol$^{-1}$ nm$^{-2}$. Configurations were extracted from the pulling simulations at 0.1 nm intervals until the C5aR–PMX53 COM–COM distance was 3.0 Å, and at 0.2 nm intervals thereafter until the final COM–COM distance was 6.0 Å. In total, 36 configurations were generated to serve as umbrella sampling windows. Each of the 36 umbrella sampling windows were equilibrated for 100 ps in an NPT ensemble followed by a 40 ns production run while applying a 1000 kJ mol$^{-1}$ nm$^{-2}$ force constant along the z-axis on the PMX53 molecule. Finally, the free energy profile of transferring PMX53 from its bound state to an unbound state was calculated using the weighted histogram analysis method (WHAM) as implemented in GROMACS v5.1.2. Bootstrap analysis was used for estimation of statistical errors.

**Analysis of non-covalent interactions**. The various non-covalent interactions were estimated using built-in GROMACS tools and in-house Perl scripts. Hydrogen bonds were estimated using the gmx hbond tool using default criteria. Cation-π and salt-bridge interactions were defined based on the distance criteria described elsewhere.

**Statistics and reproducibility**. Data presented in this study are expressed as mean ± standard error of replicate measurements and the number of replicates is specified in figure legends.

## Data availability
The source data generated and/or analyzed in the current study are included in this article as Supplementary Data 1, which includes those for Figs. 1d, e, 2i–k, 3h–j, 5a–d, S2a, b and S7. All other data that support the findings of the present study are available from the corresponding author upon request.

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

## Acknowledgements

This work was supported by the Fonds National de la Recherche Scientifique (F.R.S.-FNRS grant numbers: PDR T. 0070.16 to D.A.), the Research Department of the Communauté française de Belgique (Concerted Research Action), the Université catholique de Louvain (Fonds Spéciaux de Recherche), the 'MOVE-IN Louvain' Incoming post-doc Fellowship programme to A.C.D., and the National Institute of Health of the United States (R35GM128641 to C.Z.). D.A. is Research Associate, and A.C.D and M.K. are postdoctoral fellows at the FNRS. H.F. gratefully acknowledges financial support from Biomedical Research Council of A*STAR. This work was also supported by funding from National Institute of Health. The computational work was performed on resources of the National Supercomputing Centre, Singapore (https://www.nscc.sg). Figure cartoons created with BioRender.com.

## Author contributions

A.C.D. and M.K. set up and performed the A.F.M. experiments. D.A. and A.C.D. coanalyzed the experimental data and performed calculations. C.Z. and H.L. provided some of the ligands and cloned, purified, and reconstituted C5aR. H.F. and R.N.V.K.D. set up and performed the MD/SMD simulations and analysis. All authors wrote the paper.

## Competing interests

The authors declare no competing interests.
