## [Peer Review File · Communications Biology]

Reviewers' Comments:

Reviewer #1:

Remarks to the Author:

This paper by Dumitru et al., probes the ligand binding and activation of the human C5aR1 using a combination of AFM, MD simulation and some biochemical studies. Using the natural agonist C5a, a synthetic antagonist, and a couple of different receptor mutants, the authors propose a positive allosteric mechanism regulating the two-site binding of C5a to the receptor. Overall, the study is beautifully design, carefully performed, and well presented to researchers interested in C5aR biology as well as GPCR biology in general. While we have witnessed a surge of structural data on GPCRs, which the authors also acknowledge in their manuscript, these kinds of studies probing the dynamic aspects of ligand-receptor interaction, and potentially, receptor-effector interaction, are invaluable to develop a better understanding of GPCR activation and signaling. Thus, I strongly recommend publication of this study in *Communications Biology*. I have a couple of rather cosmetic comments, which the authors may consider addressing by textual revision.

1. The interaction of chemokines with their cognate GPCRs has also been proposed to involve a similar two-site binding mechanisms. Authors may consider extrapolating the utility of their approach to chemokine receptors in the discussion section.
2. The second C5a receptor, C5aR2 (C5L2), displays an interesting functional divergence compared to C5aR1 (recent references: doi: 10.1016/j.tibs.2020.04.004 and 10.1074/jbc.RA119.007485). It would be interesting to bring this up briefly in the discussion, perhaps to speculate about potential differences agonist-receptor interaction.
3. Authors may consider including the Gi-coupling data for the Y-F mutant of C5aR1, if readily available. If they observe a difference in EC50, this may further support their notion of allosteric mechanism between the BS and ES sites. If this data is not available, or not possible to easily generate considering the lock-down situation, authors may consider citing available literature, if any, or at least mention this possibility for future studies.
4. Line 258 – “We performed several functional assays with the C5aR and the two mutants and observed”. The only functional assay presented in the manuscript is Gi-coupling using GTP-gamma-S. Perhaps, the authors mean multiple replicates, and they should revise it accordingly.
5. It may be useful to measure, or at least discuss, how the two mutations (i.e. RY-VV and D-A) influence the binding affinity of C5a (and PMX53). This can further complement the AFM and MD simulation data.

Arun K. Shukla, Ph.D.

Reviewer #2:

Remarks to the Author:

This is an elegant and nicely presented manuscript describing the application of atomic force microscopy (AFM) to measuring the thermodynamics and kinetics of interactions of C5a receptor (C5aR) with a small molecule ligand and with its natural ligand C5a. There are two key advances. First, the establishment of this technology will enable detailed studies of interactions between many GPCRs and their natural or pharmacological binding partners. Second, since C5a has previously been shown to interact with two distinct regions of C5aR, the authors determine the contributions of each region to the binding thermodynamics and dissociation kinetics, enhancing our understanding of the binding mechanism.

Major Comments

- Page 9, C5aRR175/Y258 dictates the binding kinetics. This double-mutation appears to substantially

increase the dissociation rate without affect the equilibrium constant. This can only occur if the association rate is also increased. Do the authors have any data on the rate of association or can they, at least, suggest why this would be increased by this mutation?

- Lines 323-328. It is not obvious to me that mutation of these two tyrosine residues alone would completely abolish interactions at the BS. A more convincing mutation would be deletion of the N-terminal region.
- The finding that binding to the two regions (BS and ES) is cooperative/allosteric is not unexpected. In fact, this is expected for any intermolecular binding event involving two separate interacting regions due to the decrease in translational and rotational entropy when the first interaction forms (see Jencks PNAS, 1981, 78, 4046-4050).
- It is not clear to me whether the state of C5a being studied here is the active or inactive conformation, or perhaps different conformations in different experiments. The authors should clarify this. It also has a bearing on the free energy diagram (Fig 6), which only shows one fully-bound form, rather than both inactive and activated states. As a possible way to incorporate both forms, see a recent paper on chemokine-receptor interactions: Sanchez et al., J. Biol. Chem. (2019) 294, 3464–3475.

Minor Comments

- Page 5-6. The data indicate that the C5a receptor protrudes substantially further from the membrane surface on the C-terminal side than on the N-terminal side. Is this consistent with predictions from the 3D structure of the receptor?
- Fig 1C, legend refers to an inset but this is not shown in the image.
- Fig S4B. It is quite difficult to identify the regions corresponding to each ECL and ICL in this figure. It would be helpful to label these explicitly.
- Line 214. From Fig S4B, it appears that the termini of TM5 and TM7 are quite flexible.
- Fig 4A legend "Conformations of PMX53 at t = 0 ps, t = 500 ps, and t = 100 ps derived from the COM pulling simulation". Should this be "Positions of PMX53 at t = 0 ps, t = 100 ps, and t = 500 ps derived from the COM pulling simulation"?
- Fig 4B-F: the resolution needs to be improved.
- Lines 238-240. Fig 4C does not show a loss of intermolecular interactions at 308 ps. Should this be Fig 4E?
- Line 244. Fig 4F and J?
- Lines 319 and 330. The K_d values given do not seem to correspond correctly to the ΔG_{bu} values determined from the data. ΔG_{bu} = -3.9 kcal/mol should give a K_d in the low mM range (not 0.8 M). ΔG_{bu} = -4.7 kcal/mol should give a K_d slightly below 1 mM (not 20 mM). These changes will not affect any conclusion of the paper.
- Fig 5 legend. C and D need to be swapped.

- Fig 6. I suggest adding to the legend that the vertical axis is not to scale. Alternatively, the figure could be made to scale, which would emphasize the point about cooperativity between binding sites.

Manuscript COMMSBIO-20-2481 "Submolecular probing of the complement C5a receptor-ligand binding reveals a cooperative two-site binding mechanism" Dumitru *et al.*

Point-by-Point Response to the Reviewers

Comments Reviewer #1

Reviewer #1: This paper by Dumitru et al., probes the ligand binding and activation of the human C5aR1 using a combination of AFM, MD simulation and some biochemical studies. Using the natural agonist C5a, a synthetic antagonist, and a couple of different receptor mutants, the authors propose a positive allosteric mechanism regulating the two-site binding of C5a to the receptor. Overall, the study is beautifully design, carefully performed, and well presented to researchers interested in C5aR biology as well as GPCR biology in general. While we have witnessed a surge of structural data on GPCRs, which the authors also acknowledge in their manuscript, these kinds of studies probing the dynamic aspects of ligand-receptor interaction, and potentially, receptor-effector interaction, are invaluable to develop a better understanding of GPCR activation and signaling. Thus, I strongly recommend publication of this study in Communications Biology. I have a couple of rather cosmetic comments, which the authors may consider addressing by textual revision.

Authors: We are happy to hear that our work was received with enthusiasm and we thank the reviewer for the encouraging comments. Below we have explained point-by-point how we have addressed your questions to strengthen our manuscript.

Reviewer #1: 1. The interaction of chemokines with their cognate GPCRs has also been proposed to involve a similar two-site binding mechanisms. Authors may consider extrapolating the utility of their approach to chemokine receptors in the discussion section.

Authors: Indeed, our approach could be useful for a better understanding of mechanistic insights into chemokine receptors binding and activation. We have now updated the discussion accordingly (lines 421-424).

Reviewer #1: 2. The second C5a receptor, C5aR2 (C5L2), displays an interesting functional divergence compared to C5aR1 (recent references: doi: 10.1016/j.tibs.2020.04.004 and 10.1074/jbc.RA119.007485). It would be interesting to bring this up briefly in the discussion, perhaps to speculate about potential differences agonist-receptor interaction.

Authors: Thank you for this kind suggestion. We have now updated the discussion about how our findings could also provide more insights into the functionally divergent roles of C5aR1 and C5aR2 receptors (lines 414-421).

Reviewer #1: 3. Authors may consider including the Gi-coupling data for the Y-F mutant of C5aR1, if readily available. If they observe a difference in EC50, this may further support their notion of allosteric mechanism between the BS and ES sites. If this data is not available, or not possible to easily generate considering the lock-down situation, authors may consider citing available literature, if any, or at least mention this possibility for future studies.

Authors: We thank the reviewer for the suggestion. Previous studies have shown that mutations of Y11F and Y14F in C5aR1 could greatly compromise receptor signaling induced by C5a, but not the small agonist peptide-induced signaling¹. This data supports the allosteric mechanism between the BS and ES sites. We cited this study in our revised manuscript (lines 357-358).

Reviewer #1: 4. Line 258 – “We performed several functional assays with the C5aR and the two mutants and observed”. The only functional assay presented in the manuscript is Gi-coupling using GTP-gamma-S. Perhaps, the authors mean multiple replicates, and they should revise it accordingly.

Authors: Thank you for highlighting this error. We have revised the text in the main manuscript accordingly (line 265). We also added the number of replicates in the caption of Fig S7.

Reviewer #1: 5. It may be useful to measure, or at least discuss, how the two mutations (i.e. RY-VV and D-A) influence the binding affinity of C5a (and PMX53). This can further complement the AFM and MD simulation data.

Authors: We thank the reviewer for the suggestion. We do not have access to radioactive C5a or PMX53. It would be very difficult for us to measure the affinity of C5a or PMX53 for the C5a receptor. Our MD and SMD simulation studies suggested critical roles of R175, D282 and Y258 in the ES for peptide ligand-binding. Our evidences strongly suggest that the R6-D282 salt-bridge is the most critical interaction for ligand binding. Following formation of the R6-D282 interaction, W5 and R6 rearrange to saddle Y258. We propose that the R6-D282 salt-bridge and R6-Y258-W6 charged/stacking interaction act as anchors while the rest of PMX53 molecule samples conformations that allow for favorable PMX53-C5aR binding. The subsequent establishment of the hydrogen bonding network between PMX53's cyclic backbone and ECL2 residues (sidechain of R175 and backbone of other ECL2 residues), as observed in the crystal structure, could in turn complete the binding process. Such a model suggests that D282 is critical for the initial binding while the residence time of the bound ligand is dictated by interactions with Y258 and ECL2 residues, and is in agreement with our findings.

Functional assays confirmed that mutations of R175, D282 and Y258 residues led to significantly compromised signaling of C5aR induced by C5a (Fig. S7). This is also consistent with previously reported crystal structures of C5aR with PMX53 (PDB IDs 6C1Q and 6C1R), in which all three residues form direct interactions with PMX53. PMX53 was developed based on the C-terminal peptide of C5a. It is likely that the C-terminal peptide of C5a forms similar interactions with C5aR and mutations of R175, D282 and Y258 can therefore weaken C5a

binding. We have added discussion of these three mutations in the section 'MD simulations identify key residues involved in antagonist binding to C5aR' stating that they likely weaken the interactions between the peptide ligands and C5aR (Line 262). We also cited a previous study showing a critical role of D282 in C5a-induced C5aR signaling in this section.

Reviewer #2

Reviewer #2: This is an elegant and nicely presented manuscript describing the application of atomic force microscopy (AFM) to measuring the thermodynamics and kinetics of interactions of C5a receptor (C5aR) with a small molecule ligand and with its natural ligand C5a. There are two key advances. First, the establishment of this technology will enable detailed studies of interactions between many GPCRs and their natural or pharmacological binding partners. Second, since C5a has previously been shown to interact with two distinct regions of C5aR, the authors determine the contributions of each region to the binding thermodynamics and dissociation kinetics, enhancing our understanding of the binding mechanism.

Authors: Thank you for your encouraging and constructive comments. Below we have explained point-by-point how we have addressed your questions to strengthen our manuscript.

Reviewer #2: 1. Page 9, C5aRR175/Y258 dictates the binding kinetics. This double-mutation appears to substantially increase the dissociation rate without affect the equilibrium constant. This can only occur if the association rate is also increased. Do the authors have any data on the rate of association or can they, at least, suggest why this would be increased by this mutation?

Authors: This is an interesting question. With the FNdY model we can also gain access to the k_{on} , that only suggests a small increase of the rate of association from $1.69 \times 10^3 \text{ M}^{-1} \text{ S}^{-1}$ for the C5aR^{WT} to $1.74 \times 10^3 \text{ M}^{-1} \text{ S}^{-1}$ for the C5aR^{R175V/Y258H} mutant. As we mentioned in the paper, this phenomenon is also observed in MD simulation for the PMX53 with the C5aR and C5aR^{R175V/Y258H}. MD revealed a reduction in the number of hydrogen bonds formed between C5aR^{R175V/Y258H} and PMX53, particularly involving residues from ECL2. The R175V mutation causes a 66% reduction in the number of hydrogen bonds formed between ECL2 and PMX53 (1.57 ± 0.91) as compared to the wild type (4.62 ± 0.93 ; **Fig S5B**). The PMF profile for C5aR^{R175V/Y258H} shows a significant drop ($\sim 43\%$) in the height of the energy barrier crossed during PMX53 dissociation ($-12.2 \text{ kcal mol}^{-1}$ for double-mutant vs. $-21.5 \text{ kcal mol}^{-1}$ for WT), although resulting in a slight reduction ($\sim 12\%$) in ΔG_{bu} ($-12.2 \text{ kcal mol}^{-1}$ for C5aR^{R175V/Y258H} vs. $-13.8 \text{ kcal mol}^{-1}$ for C5aR^{WT}) (**Fig S6F**). Results from our MD and SMD studies are in good agreement with experimental data obtained by AFM where we observed a slight decrease in the ΔG_{bu} ($\sim 3\%$) but a much important reduction in residence time ($\sim 40\%$) that can be directly linked with the reduction of the height of the energy barrier crossed during PMX53 dissociation.

Reviewer #2: 2. Lines 323-328. It is not obvious to me that mutation of these two tyrosine residues alone would completely abolish interactions at the BS. A more convincing mutation would be deletion of the N-terminal region.

Authors: Indeed, the deletion of the N-terminal region of C5aR appears as the most straightforward control to abolish the interactions at the BS. However, several attempts at obtaining N-terminal truncated C5aR led to the conclusion that protein folding and insertion in the lipid bilayer are perturbed by this mutation. Amino-terminal sulfation of tyrosine residues C5aR contributes to formation of the docking site for the C5a anaphylatoxin², so we used this approach to reduce the binding affinity between the C5a rigid core and receptor BS.

Reviewer #2: 3. The finding that binding to the two regions (BS and ES) is cooperative/allosteric is not unexpected. In fact, this is expected for any intermolecular binding event involving two separate interacting regions due to the decrease in translational and rotational entropy when the first interaction forms (see Jencks PNAS, 1981, 78, 40464050)

Authors: We thank the reviewer for pointing this out and we updated the manuscript connecting our results to this previous work. (Lines 352-355)

Reviewer #2: 4. It is not clear to me whether the state of C5a being studied here is the active or inactive conformation, or perhaps different conformations in different experiments. The authors should clarify this. It also has a bearing on the free energy diagram (Fig 6), which only shows one fully-bound form, rather than both inactive and activated states. As a possible way to incorporate both forms, see a recent paper on chemokine-receptor interactions: Sanchez et al., J. Biol. Chem. (2019) 294, 3464–3475.

Authors: Thank you for your very interesting comment. The C5aR was purified and reconstituted into liposomes without any ligand and G proteins. Therefore, it represented an apo state, or a low agonist affinity state. It has been suggested that GPCR agonists alone cannot stabilize receptors in active conformational states or high agonist affinity states^{3,4}. Within the FD-based AFM experiments, the goal is to study the C5a binding to the C5aR alone in its apo state. Without G proteins to stabilize the active receptor conformation, our experiments did not probe the receptor activation event. This is why we only presented a free energy diagram related to the ligand binding mechanism without any link to the receptor activation, as this is beyond the scope of our study.

Reviewer #2: 5. Page 5-6. The data indicate that the C5a receptor protrudes substantially further from the membrane surface on the C-terminal side than on the N-terminal side. Is this consistent with predictions from the 3D structure of the receptor?

Authors: The referee asked for the consistency of the C5aR protrusion from the membrane between our AFM experiments and the 3D structure. However, this comparison is not straightforward, as the crystal structure only partially resolved the structure of the loops and N-/C- termini. In addition, the AFM topography image is obtained under an applied force (even

though we tried to maintain it as low as possible), which can partially deform flexible regions of the receptors. Nevertheless, the obtained values are in good agreement with height data previously obtained by AFM for other GPCRs^{5,6}.

Reviewer #2: 6. Fig 1C, legend refers to an inset but this is not shown in the image.

Authors: We apologize for the error. The inset corresponds to the panel 1D. This was corrected in the figure caption.

Reviewer #2: 7. Fig S4B. It is quite difficult to identify the regions corresponding to each ECL and ICL in this figure. It would be helpful to label these explicitly.

Authors: We thank the reviewer for the suggestion. We have now modified Fig. S4B.

Reviewer #2: 8. Line 214. From Fig S4B, it appears that the termini of TM5 and TM7 are quite flexible.

Authors: Thank you for highlighting this error. Indeed the sentence should be "... most of the structural fluctuations were observed primarily in the second extracellular loop (ECL2) and the terminal regions of TM5, TM6 and TM7, while the 7-TM core of the protein remained fairly stable (Fig S4B)". This is corrected in the manuscript (lines 215-217).

Reviewer #2: 9. Fig 4A legend "Conformations of PMX53 at t = 0 ps, t = 500 ps, and t = 100 ps derived from the COM pulling simulation". Should this be "Positions of PMX53 at t = 0 ps, t = 100 ps, and t = 500 ps derived from the COM pulling simulation"?

Authors: Thank you for highlighting this error. Indeed the sentence should be "Conformations of PMX53 at t = 0 ps, t = 500 ps, and t = 1000 ps derived from the COM pulling simulation". This is corrected in the Fig 4A legend.

Reviewer #2: 10. Fig 4B-F: the resolution needs to be improved.

Authors: Thank you. We now provided a Figure 4 with a higher resolution and increased the size of the labels.

Reviewer #2: 11. Lines 238-240. Fig 4C does not show a loss of intermolecular interactions at 308 ps. Should this be Fig 4E?

Authors: Indeed, we now corrected it into Figure 4E.

Reviewer #2: 12. Line 244. Fig 4F and J?

Authors: Thank you. It is now corrected

Reviewer #2: 13. Lines 319 and 330. The K_d values given do not seem to correspond correctly to the ΔG_{bu} values determined from the data. $\Delta G_{bu} = -3.9$ kcal/mol should give a K_d in the low mM range (not 0.8 M). $\Delta G_{bu} = -4.7$ kcal/mol should give a K_d slightly below 1 mM (not 20 mM). These changes will not affect any conclusion of the paper.

Authors: We apologize for the confusion. In order to compare the value obtained by our AFM experiments with value obtained in bulk assays, we need to compare the method assays parameters [F_{eq} , $k_{off}(F_{eq})$] to bulk assays parameters which are the unstressed values (k_{off} , k_{on}) for both off rate and on rate respectively. Although the dissociation constant $K_D = k_{off}/k_{on}$ is a measure of the binding free energy ΔG_{bu} , the relationship is predicted on kinetics of a first order reaction in solution, $\Delta G_{bu} = k_b T \ln(K_D/55.6 \text{ M/L})$ with $1/55.6 = 0.018 \text{ (L/M)}$ being the partial molar volume of water. If the partial molar volume of water is left out, the comparison is not accurate even though consistent with chemistry.

Reviewer #2: 14. Fig 5 legend. C and D need to be swapped.

Authors: Thank you. We corrected this in the manuscript.

Reviewer #2: 15. Fig 6. I suggest adding to the legend that the vertical axis is not to scale. Alternatively, the figure could be made to scale, which would emphasize the point about cooperativity between binding sites.

Authors: We thank the reviewer for the suggestion. The vertical axis was resized and it is now made to scale, which indeed emphasizes better the cooperativity between the two binding sites.

References

- 1 Farzan, M. *et al.* Sulfated Tyrosines Contribute to the Formation of the C5a Docking Site of the Human C5a Anaphylatoxin Receptor. *The Journal of Experimental Medicine* **193**, 1059-1066, doi:10.1084/jem.193.9.1059 (2001).
- 2 Farzan, M. *et al.* Sulfated Tyrosines Contribute to the Formation of the C5a Docking Site of the Human C5a Anaphylatoxin Receptor. *Journal of Experimental Medicine* **193**, 1059-1066 (2001).
- 3 Rosenbaum, D. M. *et al.* Structure and function of an irreversible agonist- β_2 adrenoceptor complex. *Nature* **469**, 236-240 (2011).
- 4 Sounier, R. *et al.* Propagation of conformational changes during μ -opioid receptor activation. *Nature* **524**, 375-378 (2015).
- 5 Alsteens, D. *et al.* Imaging G protein-coupled receptors while quantifying their ligand-binding free-energy landscape. *Nature Methods* **12**, 845-851 (2015).
- 6 Lo Giudice, C., Zhang, H., Wu, B. & Alsteens, D. Mechanochemical Activation of Class-B G-Protein-Coupled Receptor upon Peptide-Ligand Binding. *Nano Letters* **20**, 5575-5582 (2020).

REVIEWERS' COMMENTS:

Reviewer #1 (Remarks to the Author):

The authors have addressed my comments reasonably well, and the revised manuscript is ready for publication. I congratulate the authors for this wonderful piece of work.

Arun K. Shukla, Ph.D.

Reviewer #2 (Remarks to the Author):

The authors have appropriately addressed the previous comments.